# The modelled climatic response to the 18.6-year lunar nodal cycle and its role in decadal temperature trends

Manoj Joshi[1, 2], Rob A. Hall[1], David P. Stevens[3], Ed Hawkins[4]

[1]School of Environmental Sciences, University of East Anglia, Norwich NR4 7TJ, United Kingdom

[2]Climatic Research Unit, University of East Anglia, Norwich NR4 7TJ, United Kingdom

[3]School of Mathematics, University of East Anglia, Norwich NR4 7TJ, United Kingdom

[4]National Centre for Atmospheric Science, Department of Meteorology, University of Reading, Reading RG6 6BB, United Kingdom

*Correspondence to:* Manoj Joshi (m.joshi@uea.ac.uk)

**Abstract**. The 18.6-year lunar nodal cycle arises from variations in the angle of the Moon's orbital plane. Previous work has linked the nodal cycle to climate but has been limited, either by the length of observations analysed, or geographical regions considered in model simulations of the pre-industrial period. Here we examine the global effect of the lunar nodal cycle in multi-centennial climate model simulations of the pre-industrial period. We find cyclic signals in global and regional surface air temperature (with amplitudes of around 0.1 K), and in ocean heat uptake and ocean heat content. The timing of anomalies of global surface air temperature and heat uptake are consistent with the so-called slowdown in global warming in the first decade of the 21st century. The lunar nodal cycle causes variations in mean sea level pressure exceeding 0.5 hPa in the Nordic seas region, thus affecting the North Atlantic Oscillation during boreal winter. Our results suggest that the contribution of the lunar nodal cycle to global temperature should be negative in the mid-2020s before becoming positive again in the early-2030s, reducing the uncertainty in time at which projected global temperature reaches 1.5C above pre-industrial levels.

## 1. Introduction

The lunar nodal cycle arises from variations in the angle of the Moon's orbital plane relative to plane of the Earth's equator (lunar declination), between 18.3° and 28.6°, over a period of 18.6 years (Pugh 1987). A potential connection of this cycle to climate is through the modulation of ocean tides (Loder and Garrett 1978), the dissipation of which are a major driver of vertical diffusion in the world's oceans (Pease et al. 1995, de Lavergne et al. 2020). The change in lunar declination results in an 18.6-year period modulation of all lunar and luni-solar tidal constituents, and potentially the resulting tidally-driven diffusion. The amplitude of the modulation varies depending on tidal constituent, but for the dominant semidiurnal and diurnal constituents ($M_2$ and $K_1$) the modulation is small (3.7% and 11.5% respectively; Table 1). Previous research has attempted to identify the effect of this signal in climate observations, but since the total modulation of the tide is small, demonstrating a significant effect on global temperature is extremely hard (Ray 2007). Regional climatic records, such as sea level in regions with large tides, or multi-century proxies, have been shown to exhibit an 18.6 year cycle (Currie et al. 1984, Yndestad et al. 2006, Yasuda et al. 2006, Gratiot et al. 2008, Agosta et al. 2013, Hamamoto and Yasuda 2021).

Modelling studies of this phenomenon are relatively rare: simple studies considering modulated stratification in the ocean have suggested an effect on global temperature (Loder and Garrett 1978). More complex studies, involving ocean circulation models (Osafune and Yasuda 2013), and most recently coupled ocean-atmosphere models, have demonstrated an effect of the nodal

cycle on the circulation of the Pacific Ocean (Osafune et al. 2020), and suggested a link to variability in the Pacific basin, particularly the Pacific Decadal Oscillation (PDO) (Tanaka et al. 2012, Osafune et al. 2014).

Here we perform millennial length runs of a coupled atmosphere-ocean global circulation model (AOGCM) to quantify the effect of a parameterisation of the lunar nodal cycle on climate. The flexibility of the parameterisation allows for sensitivity tests to be conducted. We investigate the effect of the lunar nodal cycle on long-term trends, with a particular view to understanding its role on the so-called 'slowdown' on global warming in the early part of the 21st century.

## 2. Method

Our research employs the FORTE2 climate model (Blaker et al. 2021), which uses the primitive equations of meteorology and oceanography on a sphere. The atmospheric component is the IGCM4, the ocean component is MOMA (Joshi et al. 2015, Webb 1996). The IGCM4 is run in its full 35-layer stratosphere-resolving configuration, with a horizontal resolution approximating to 2.8°. MOMA is run with a 2° horizontal resolution, and 15 vertical levels. The background vertical diffusion in the ocean component of the model, a large part of which is accounted for by tidal dissipation, is then modulated using a simple parameterisation that assumes all tidal energy is dissipated locally at the 2° grid scale.

The nodal cycle parameterisation is constructed using the geographical distribution of RMS (root mean square) current velocity magnitude for the 8 largest tidal constituents ($M_2$, $S_2$, $N_2$, $K_2$, $K_1$, $O_1$, $P_1$ and $M_f$), calculated from the TPXO7.2 inverse model (Egbert and Erofeeva 2002), multiplied by their nodal amplitudes as defined by Pugh (1987; see Table 1). The constituent-sum of modulated RMS velocity magnitude is divided by the constituent-sum of un-modulated RMS velocity magnitude, giving the relative modulation of tidal currents at each ocean gridpoint; such a normalisation is necessary because the parameterised tide in FORTE2, as in most AOGCMs, has constant amplitude in space. Note that modulations of $M_2$ and $N_2$ are 180° out of phase with the other tidal constituents, so in regions with a strongly semidiurnal tidal regime (e.g., around New Zealand) the amplitude of the nodal cycle parameterisation may be negative. $S_2$ and $P_1$ are pure solar tides so are not directly modulated by the nodal cycle, however they do contribute to total un-modulated RMS velocity magnitude and so affect the relative modulation of tidal currents.

| Constituent | Period (hours) | Typical magnitude (relative to $M_2$) | Nodal amplitude |
|---|---|---|---|
| $M_2$ | 12.42 | 1.00 | −0.037 |
| $S_2$ | 12.00 | 0.47 | 0.000 |
| $N_2$ | 12.66 | 0.19 | −0.037 |
| $K_2$ | 11.97 | 0.13 | 0.286 |
| $K_1$ | 23.93 | 0.58 | 0.115 |
| $O_1$ | 25.82 | 0.42 | 0.187 |
| $P_1$ | 24.07 | 0.19 | 0.000 |
| $M_f$ | 327.9 | 0.17 | 0.414 |

Table 1. Characteristics of the 8 tidal constituents using in the parameterisation (Pond and Pickard 1983; Pugh 1987). Subscript 2 denotes semidiurnal tides, subscript 1 denotes diurnal tides, and subscript f denotes fortnightly tides. Negative nodal amplitude indicates that the modulation of the constituent is 180° out of phase. Grey indicates a pure solar tide that is not directly modulated by the lunar nodal cycle.

The geographical shape of the function, shown in Figure 1, is determined by the relative strength of each tidal constituent at a given location and the constituent modulation amplitude. This is multiplied by a normalised 18.6-year sinusoidal cycle to yield a spatially and temporally varying modulation function. The phase of the modulation is such that, at most grid points, tidal

currents are maximum at 4.75 years into the cycle (e.g., June 2006). The Pacific and Arctic Oceans feature modulations of approximately 5% in amplitude, while the Atlantic Ocean has comparatively little modulation of tidally-driven diffusion (typically <2%). Although the tidal modulation is largest (exceeding 10%) in the Arctic and Southern Oceans, high-latitude
water columns are typically only weakly, or negatively, temperature stratified (i.e. the near-surface vertical gradient of temperature is either small or negative). So, counterintuitively, the effect of tidal modulation on climate in these regions might actually be small.

Tides are known to play a controlling role in the energetics of the global ocean, dissipating well over half of the kinetic energy in the oceans, with the greatest dissipation occurring near the ocean floor (Munk and Wunsch 1998, Egbert and Ray 2000, St.
Laurent et al. 2002). Given the uncertainties in the vertical contribution of tides to the background diffusion, two idealised perturbation runs have been performed, one in which the nodal cycle parameterisation is applied uniformly with depth to the vertical diffusion ("Constant"), and one in which it is applied such that its amplitude linearly decreases from 1 at a depth of 5000 m to 0 at the ocean surface ("Scaled"), to mirror the effect of tidal dissipation. The SCALED run should be seen as an underestimate of the near-surface effects of the lunar nodal cycle, with the CONSTANT run being an overestimate.

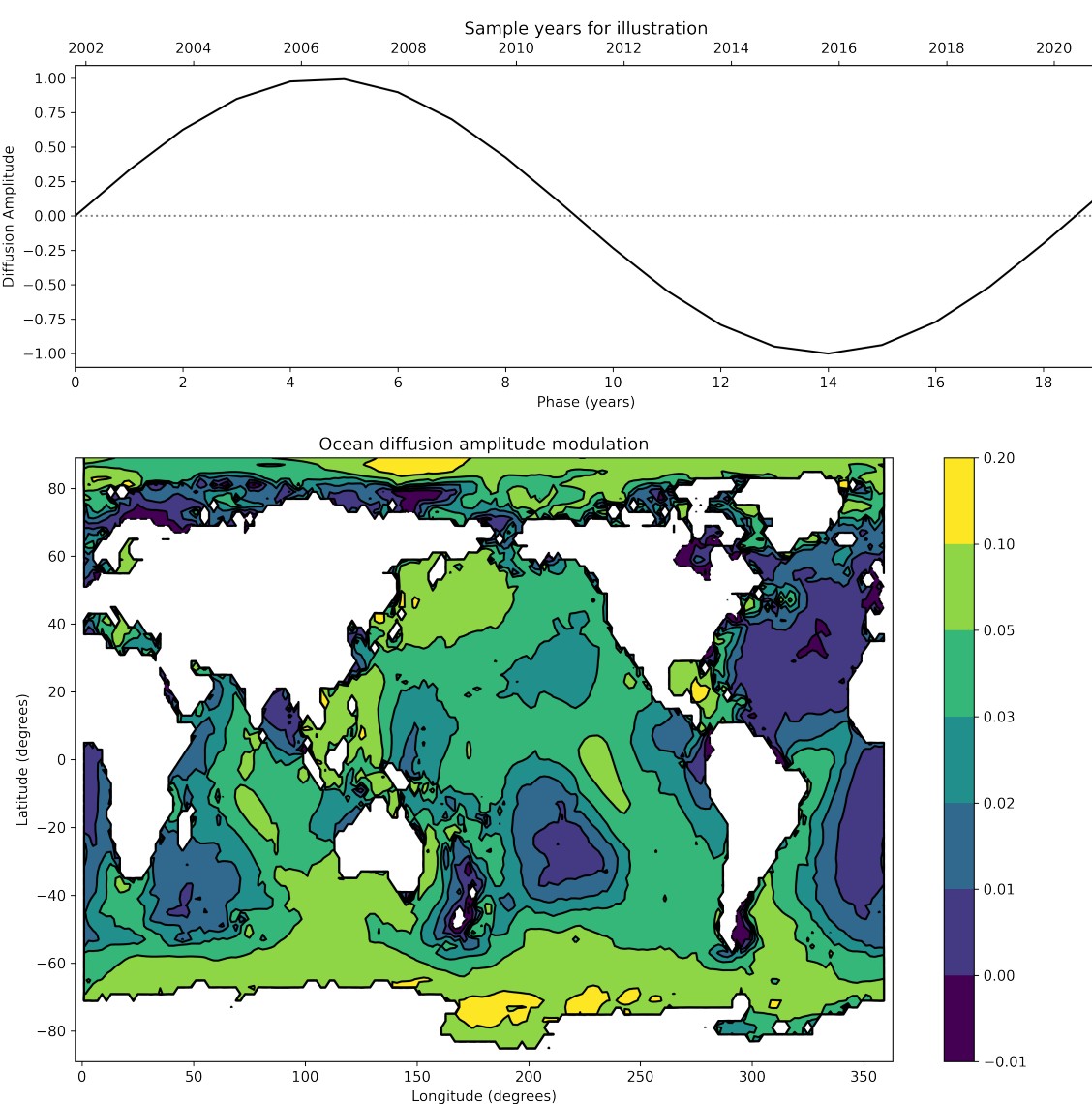


Figure 1. Top panel- variation in time of the modulation (with reference year for illustrative purposes on the top axis). The tidal modulation in the model is the top panel T(t) multiplied by the bottom panel M(x,y), multiplied by 1.0 for run CONSTANT, or a scaled function in run SCALED- see equation (1). Bottom panel- geographical distribution of the modulation of tidally-driven diffusion by the 18.6 year lunar nodal cycle.

The nodal cycle modulation is applied to the vertical diffusion with a period of 19 FORTE2 years, such that the total diffusion has the form

$$K' = K*T(t)*M(x,y)*S(z) \qquad (1)$$

where K is the standard background diffusion in FORTE2 (Blaker et al. 2021), T(t) is the sinusoidal function of Figure 1 (top

panel), M(x,y) is the geographically varying function in Figure 1 (bottom panel), and S(z) is unity for run CONSTANT, or the scaled function described above in run SCALED. Given the length of the year in FORTE2 is 360 days, such an approximation results in a nodal cycle whose length in days is within 0.7% of the observed cycle. FORTE2 is run for three configurations: pre-industrial control (as in Blaker et al. 2021), SCALED, and CONSTANT, for 2300 years, with years 1520-2280 being analysed, i.e. 760 years or 40 full cycles.

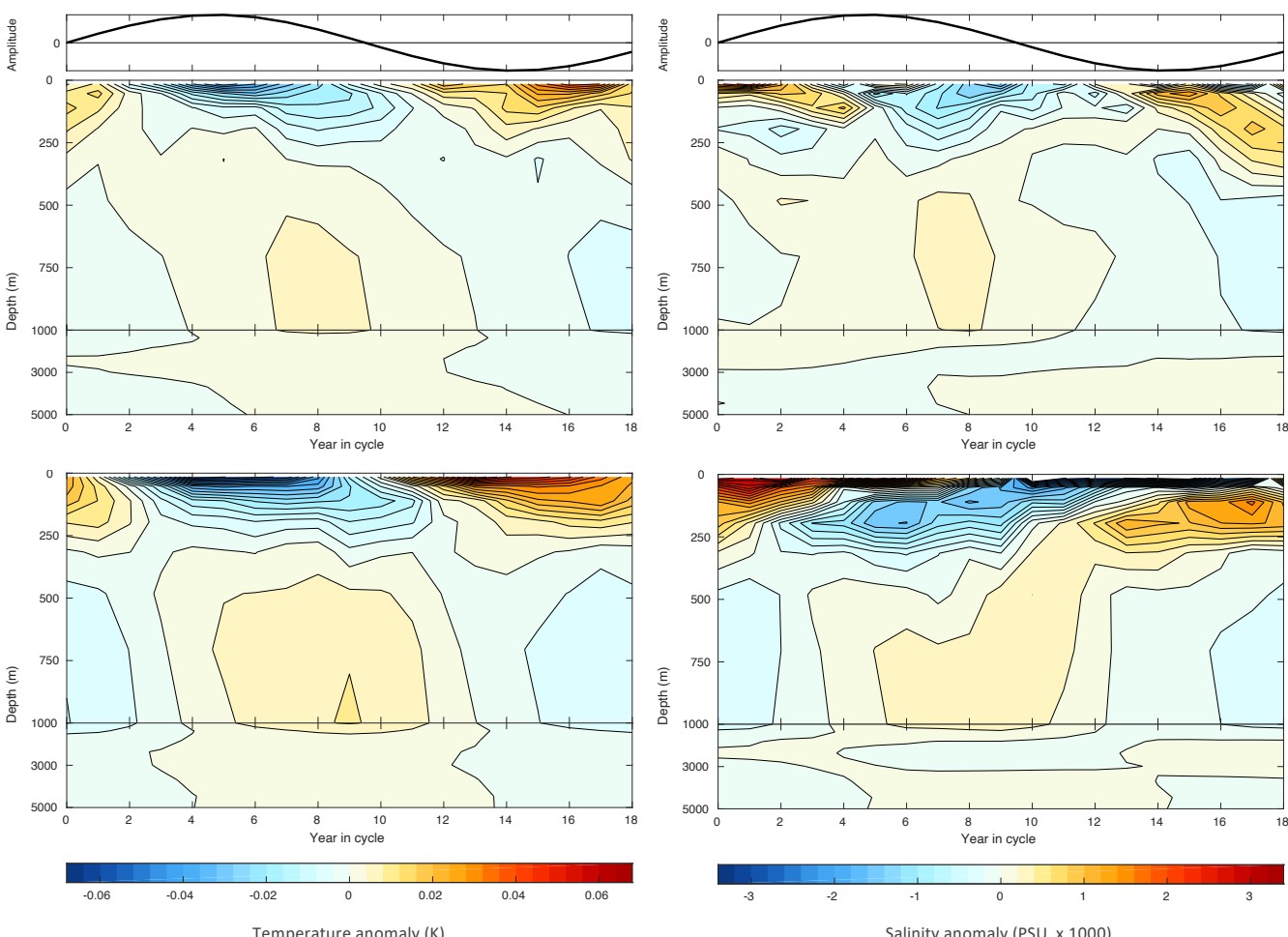

Figure 2. Top left panel- globally averaged variation of ocean temperature anomalies (K) vs ocean depth and phase (years) in the SCALED run. The phase coordinate describes the temporal modulation of the vertical diffusion shown in the inset at the top (T(t) in equation 1), and in Figure 1 (top panel), i.e. maximum diffusion is in year 5 and minimum diffusion is in year 14. Bottom left panel- as top left but for CONSTANT run. Top right panel- as top left but for global salinity (PSU). Bottom right

panel- as bottom left but for global salinity.

## 3. Results

The global averaged ocean temperature stratification has warm waters in the upper ocean and cooler waters at depth. As the amplitude of the tidally driven diffusion increases in the first phase of the nodal cycle, the global mean vertical temperature gradient is reduced with surface waters cooling and deeper waters warming. The surface temperature anomalies are larger than

those at depth, as the vertical temperature gradient is largest in the upper ocean, through the permanent thermocline. Figure 2 shows the evolution of 19-year global ocean temperature and salinity anomalies with depth as a function of the amplitude of the lunar nodal cycle diffusion modulation. In both SCALED (top left panel) and CONSTANT (bottom left panel) cases, the top 100-150 m of ocean displays a cooling in phase with maximum vertical diffusion in years 4-6. In the absence of any feedback from the atmosphere, the global mean sea surface temperature cold anomaly might be expected to peak at the same

time as the subsurface warm anomaly, which is half-way through the nodal cycle in years 9-10. This is when the tidally driven diffusion changes from its enhanced phase to a reduced phase. The response for global salinity is small. Negative salinity anomalies exist in the upper 250m of the ocean of amplitude of approximately 0.1 PSU in years 8-10 with positive anomalies in years 0-2 (Figure 2 top right and bottom right panels). The salinity anomalies below 250 m are an order of magnitude smaller than those at the surface, reflecting much smaller vertical salinity gradients are small at depth.

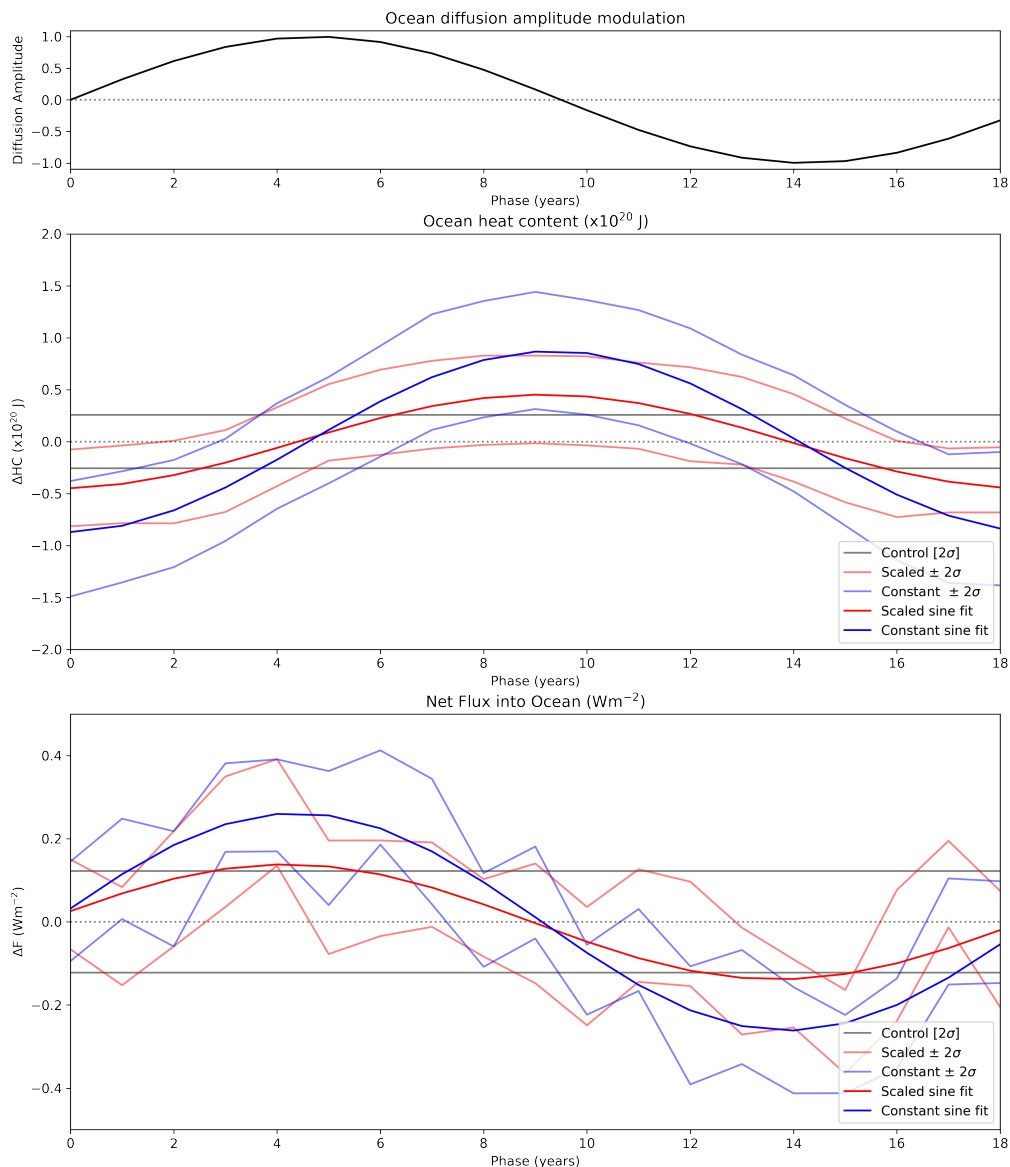


Figure 3: Top panel- variation in time of the modulation of diffusion T(t) (see equation (1). Middle panel – globally averaged ocean heat content anomaly ($10^{22}$ J) vs tidal modulation phase. The mean +/- 2 standard errors are shown for CONSTANT in thin blue, and for SCALED in thin red; sinusoidal best fit curves of global temperature anomalies versus phase are shown for CONSTANT in thick blue, and for SCALED in thick red; the mean +/- 2 standard errors of heat content anomaly in the

control integration of FORTE2 are shown for reference in black in order to demonstrate the size of the signal compared to internal variability in the control run. Bottom panel- as for middle panel, but for the globally averaged surface ocean heat flux anomaly (W m$^{-2}$) vs tidal modulation phase.

As shown in Figure 3 (bottom panel), the atmosphere almost immediately responds to the anomalously cool sea surface temperatures by fluxing heat into the ocean during years 3-7, causing an increase in total ocean heat content between years 3 and 10 (Figure 3 middle panel). The uptake of heat by the ocean results in a global ocean heat content anomaly approximately in quadrature with the surface heat flux, i.e. maximum heat content is in years 9-10 (Figure 3 middle panel), while the maximum surface flux is at years 4-5, or approximately 4.5 years or 90º out of phase with the maximum heat content. The deeper (below approximately 1000 m) temperature anomalies are largely isolated from the surface forcing and are approximately in quadrature with the nodal cycle (Figure 2). Thus the deep ocean response to the nodal cycle actually lags the response at the surface (see later). As the tidally driven diffusion reduces in the second half of the cycle, the situation described above is reversed.

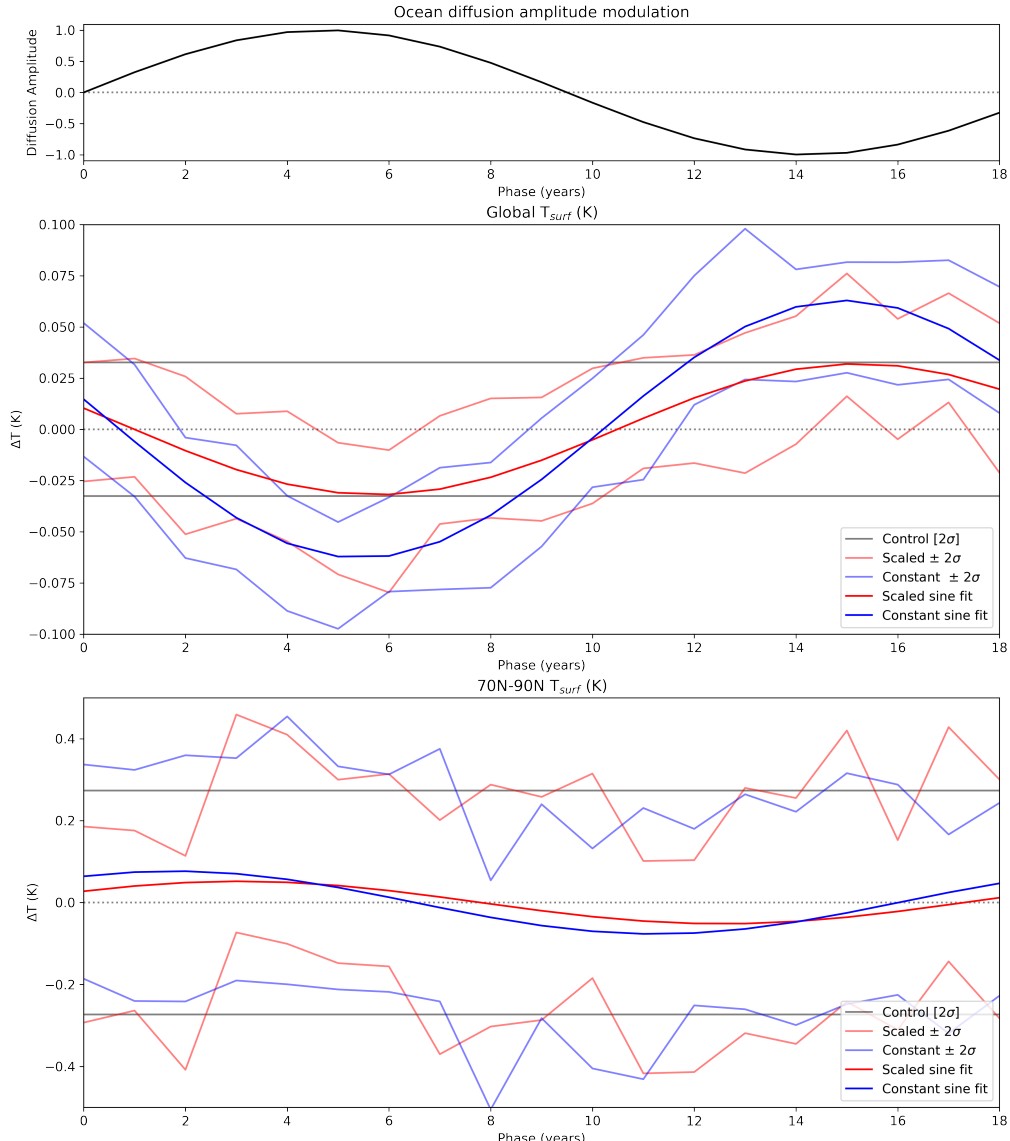

Figure 4. As Figure 3 but Middle panel- globally averaged surface temperature T$_{surf}$ (K) vs tidal modulation phase. Bottom panel- as for middle panel but for surface temperature (K) in the Arctic region (70°N-90°N).

Figure 4 shows the effect of the lunar nodal cycle on the model global mean surface temperature T$_{surf}$ expressed as a function of the phase of the cycle. A best fit of a 19-year harmonic to T$_{surf}$ shows the phases at which minimum and maximum cooling occur. Minimum global temperatures are reached within a year of the maximum diffusion occurring at year 4.5: such behaviour should be contrasted with the modelled response to transient solar or volcanic forcing, where a lag of approximately 2-3 years is present between maximum forcing and response (Gray et al. 2013). The amplitude of response in SCALED is 0.03±0.02 K while the amplitude of the response in CONSTANT is 0.06±0.02 K. The response in the SCALED run is statistically significant

only at times of max/min $T_{surf}$, while the larger response in the CONSTANT run is significant at more times. Figure 4 (bottom panel) shows the response in the Arctic region. Here the pattern is very noisy, but there is some indication of a shift in phase of temperature, which is examined in more detail below.

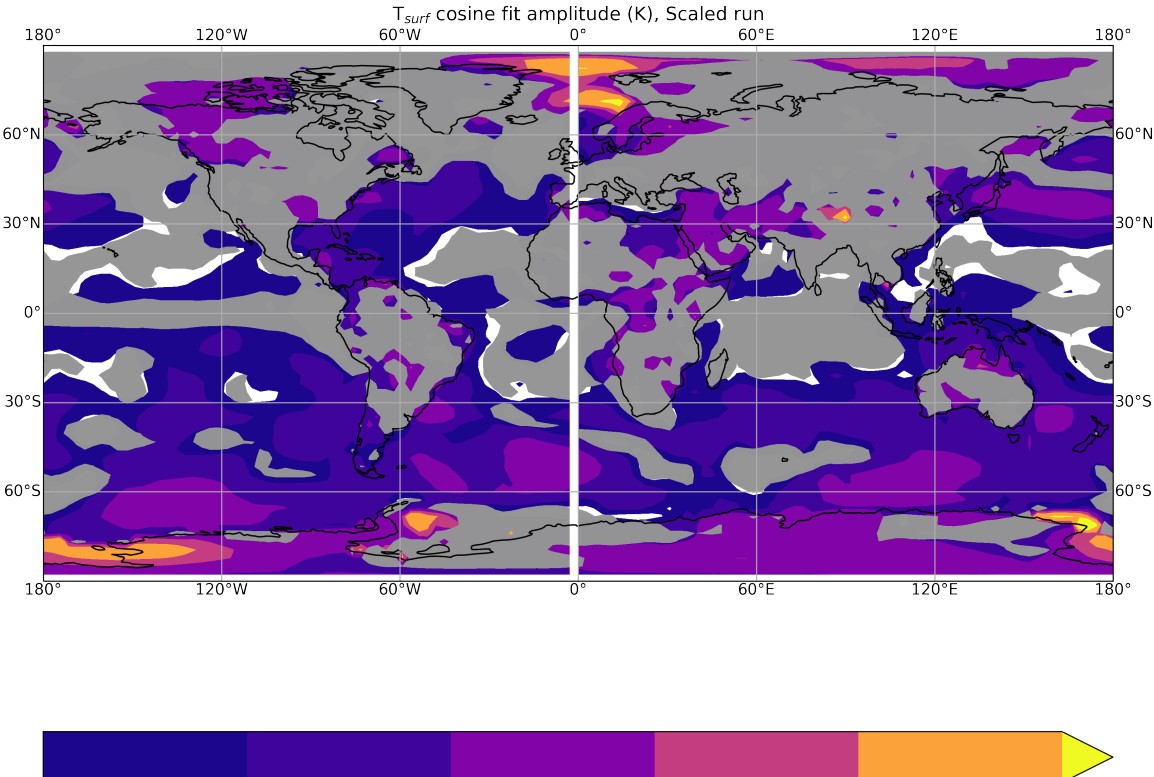

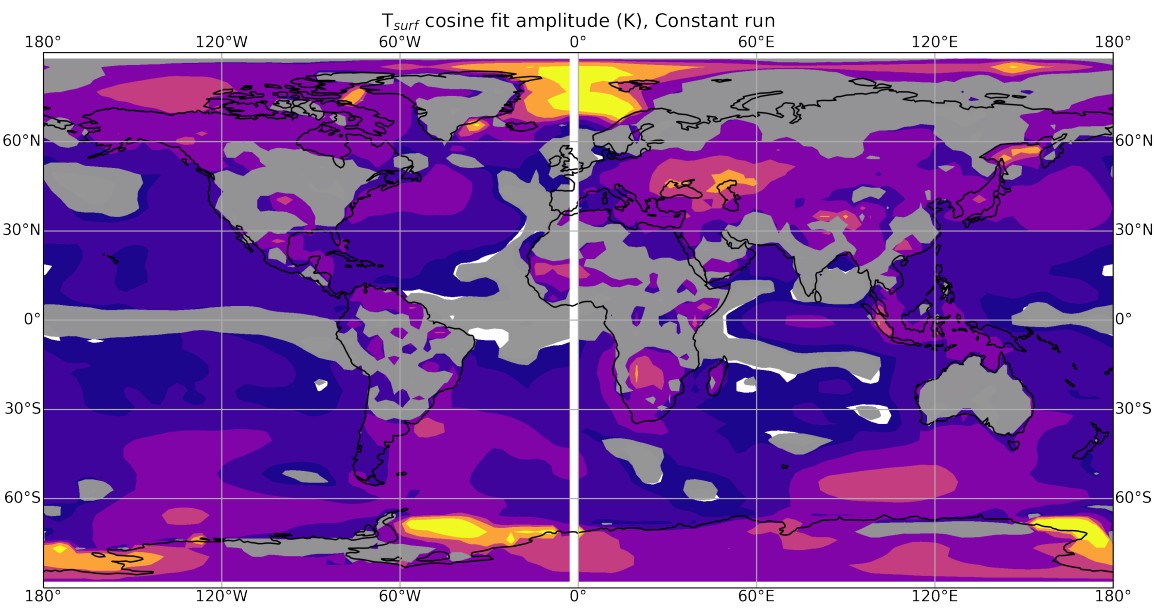

Figure 5. Top panel- geographical variation of the amplitude of the sinusoidal trigonometrical fit to surface temperature (K) whose globally-averaged counterpart is shown in Figure 4 (middle panel) in SCALED; i.e. yellow colours are where the fitted curve has an amplitude of > 0.5K. Grey shaded areas show where the amplitude is less than 2 standard errors of annually averaged $T_{surf}$ in the CONTROL integration and are used to denote areas where the response is likely to be noise; note nonlinear contour interval. Bottom panel- as top panel but for CONSTANT.


We now analyse the geographically varying response of FORTE2 to the lunar nodal cycle. Figure 5 shows the geographical variation of the amplitude of the response of $T_{surf}$ to the nodal cycle. Figure 5 exhibits quite large responses of amplitude 0.1 K in the Northwest Pacific Ocean in both SCALED (top panel) and CONSTANT (bottom panel) runs, consistent with previous work (Tanaka et al. 2012), and the Nordic seas. Generally, the response in the CONSTANT run is larger and statistically

significant in more areas than the SCALED run, consistent with the larger surface forcing of the nodal cycle in the latter. Interestingly, in both SCALED and CONSTANT runs, the subpolar North Atlantic Ocean displays a significant response of amplitude 0.05-0.1 K. The fitted response is inconsistent with the relatively small tidal forcing in the Nordic seas (Figure 1, bottom panel), and suggests a significant feedback. In addition, the CONSTANT run displays a large response on the southern flank of the Southern Ocean of amplitude > 0.3 K. Both regions are areas with significant sea-ice present, and the robustness

of the results in these areas is discussed below.

Figure 6 shows the phase in years at which minimum $T_{surf}$ is reached, with the phase being defined as in Figure 1 (top panel). Note that the globally averaged $T_{surf}$ exhibits a minimum at years 4-6 (Figure 4, middle panel). Most areas display minimum $T_{surf}$ at years 4-6 (blue-purple colors), consistent with Figure 4. A notable difference is the subpolar northeast Atlantic Ocean, much of the Arctic Ocean, and parts of the Southern Ocean, where a minimum in $T_{surf}$ occurs in years 14-18, completely out

of phase with the global response, in both SCALED and CONSTANT (red/orange colours). This polar response can be understood in terms of the local stratification: In the Nordic Seas and Southern Ocean, the ocean temperature maximum occurs at mid-depth rather than at the surface because salinity is the dominant stratifying property. As the internal ocean heat flux is upwards above the temperature maximum, increased (reduced) vertical diffusion associated with the nodal cycle leads to higher (lower) surface temperatures.

The geographically varying phases suggest a potential for geographically varying temperature and circulation responses, especially in the case of the North Atlantic Oscillation (NAO), and Southern Annular Mode (SAM), which are simulated quite well by FORTE2 (Blaker et al. 2021). Figure 7 shows the geographical variation of the amplitude of the response of November-March mean sea level pressure (MSLP) to the nodal cycle. The response is much noisier than for $T_{surf}$ (Figure 5), with most areas displaying responses that are not statistically significant (grey shaded). However, there are some regions where

significant responses do occur in both SCALED and CONSTANT runs. In particular the northeast Atlantic and Europe exhibit amplitudes exceeding 0.5 hPa, which is consistent with the large amplitude of the $T_{surf}$ response shown in Figure 5. The size of these responses in the context of variability forced by other mechanisms is discussed below. There is some indication of a wavelike response at 50S in SCALED- however, since there is little sign of such a response in CONSTANT, which has a larger $T_{surf}$ response in this region (Figure 5 bottom panel), it is unlikely that this response is robust.

Figure 8 shows the phase in years at which minimum MSLP is reached, with the phase being defined as in Figures 1 (top panel). In the Atlantic/European region, a significant signal does exist in both SCALED and CONSTANT runs, but the phasing is different. In SCALED, minimum MSLP occurs over Scandinavia in years 4-6, whereas in CONSTANT, which has a larger nodal forcing, a statistically significant minimum MSLP occurs over a wider region in years 10-14, somewhat coincident with the phase at which minimum $T_{surf}$ occurs over the North East Atlantic Ocean (Figure 6 bottom panel). The response implies

that in the opposite phase of the nodal cycle, i.e. in years 1-5, positive $T_{surf}$ anomalies over the North East Atlantic Ocean and maxima in Nov-Mar MSLP (i.e. a negative phase of the NAO) would occur, which is consistent with ideas that link loss of Arctic sea-ice with colder European land temperature in winter (Stroeve et al. 2012). We also note that in CONSTANT, the phasing in years of the minimum in $T_{surf}$ in northwest Europe is in years 4-6, almost in antiphase to the minimum in $T_{surf}$ of the adjacent North East Atlantic Ocean (see Figure 6 bottom panel).


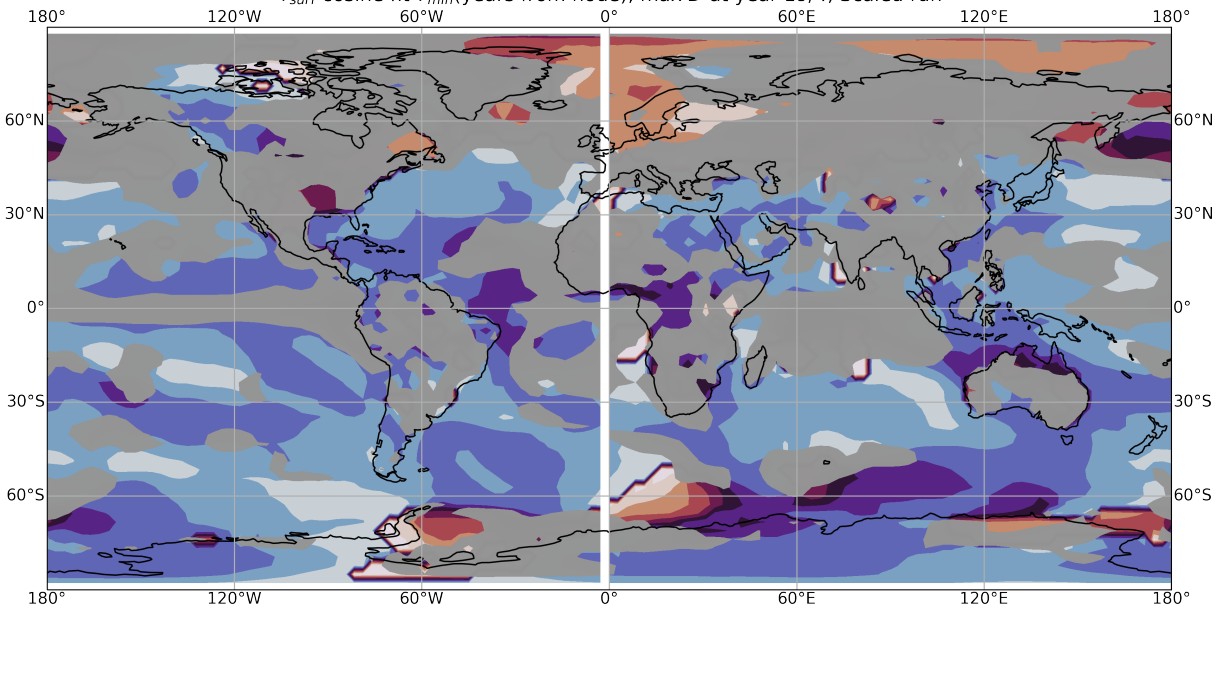

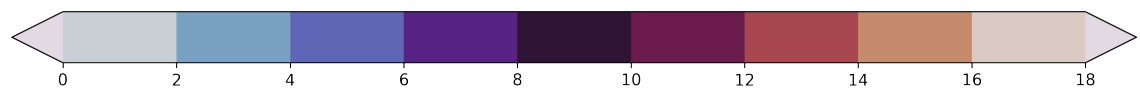

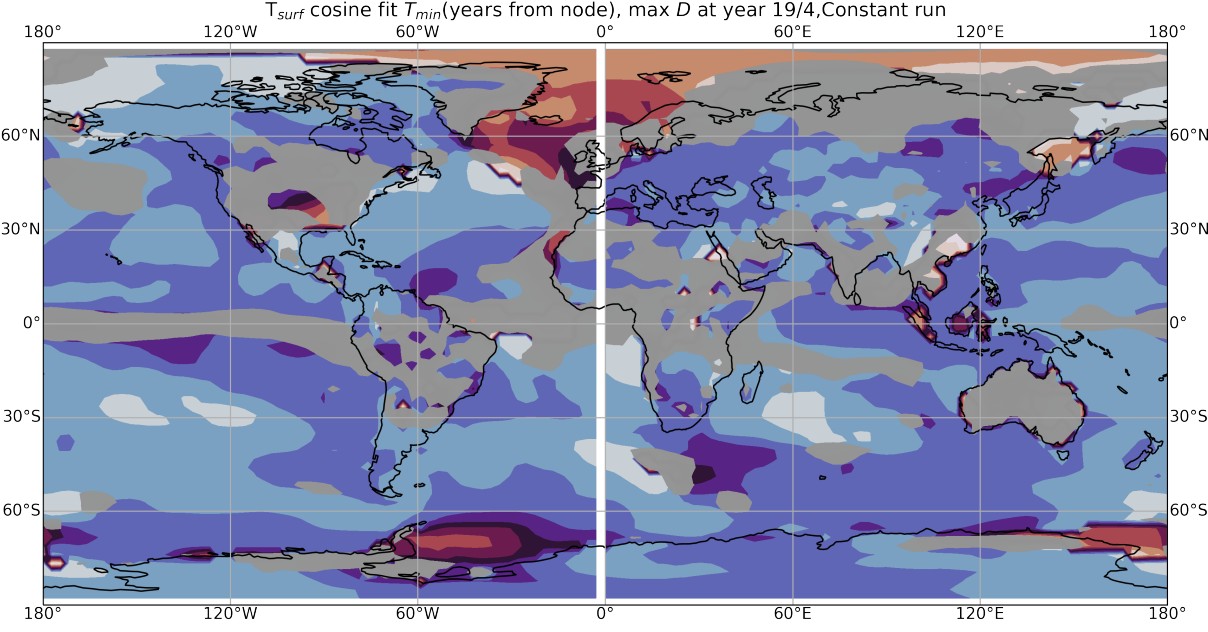

Figure 6. Top panel- geographical variation of the phase of the sinusoidal trigonometrical fit to surface temperature (K) shown in Figure 5. At each point, the colour denotes the phase in years (see Figure 1 top panel) where the fitted $T_{surf}$ reaches its most negative value. Note the cyclic colour interval, since a phase of 19 years is equivalent to a phase of 0 years. Blue colours correspond to phases associated with a globally averaged minimum in $T_{surf}$ (Figure 4 middle panel). As with Figure 5, grey shaded areas show where the amplitude is less than 2 standard errors of annually averaged $T_{surf}$ in the CONTROL integration and are used to denote areas where the response is likely to be noise. Bottom panel- as top panel but for CONSTANT.

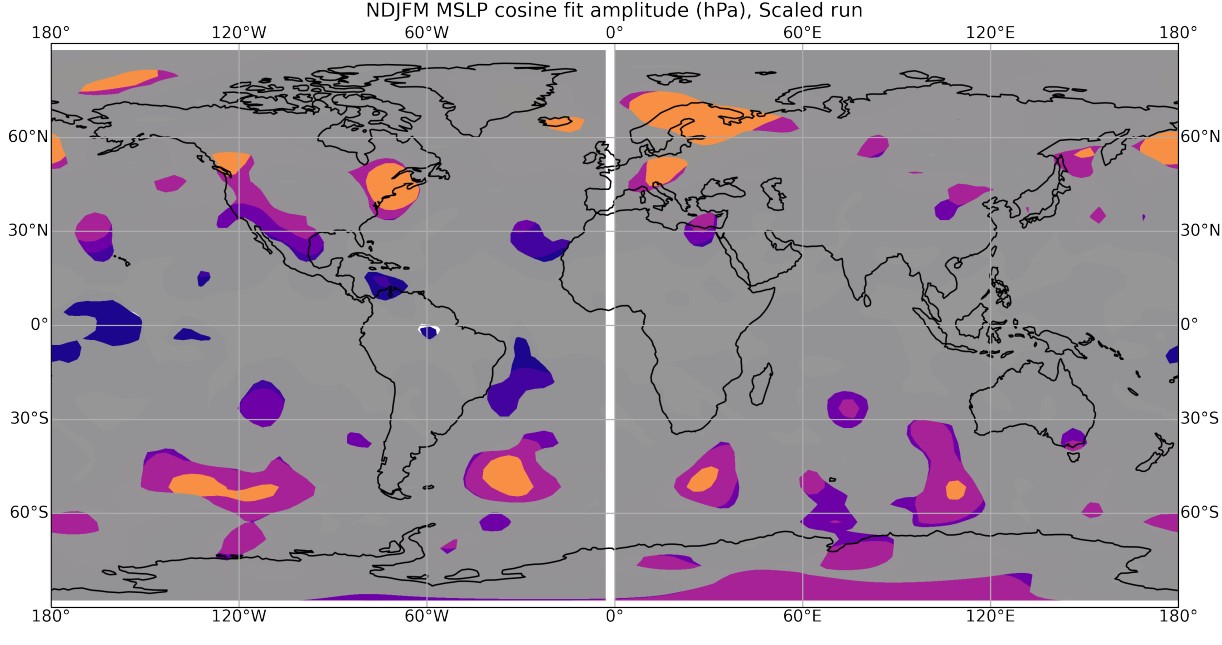

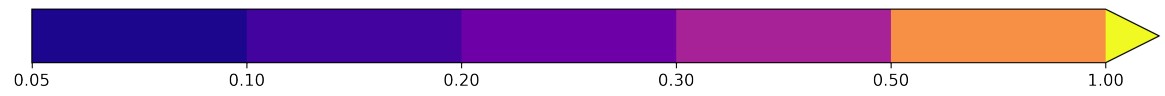

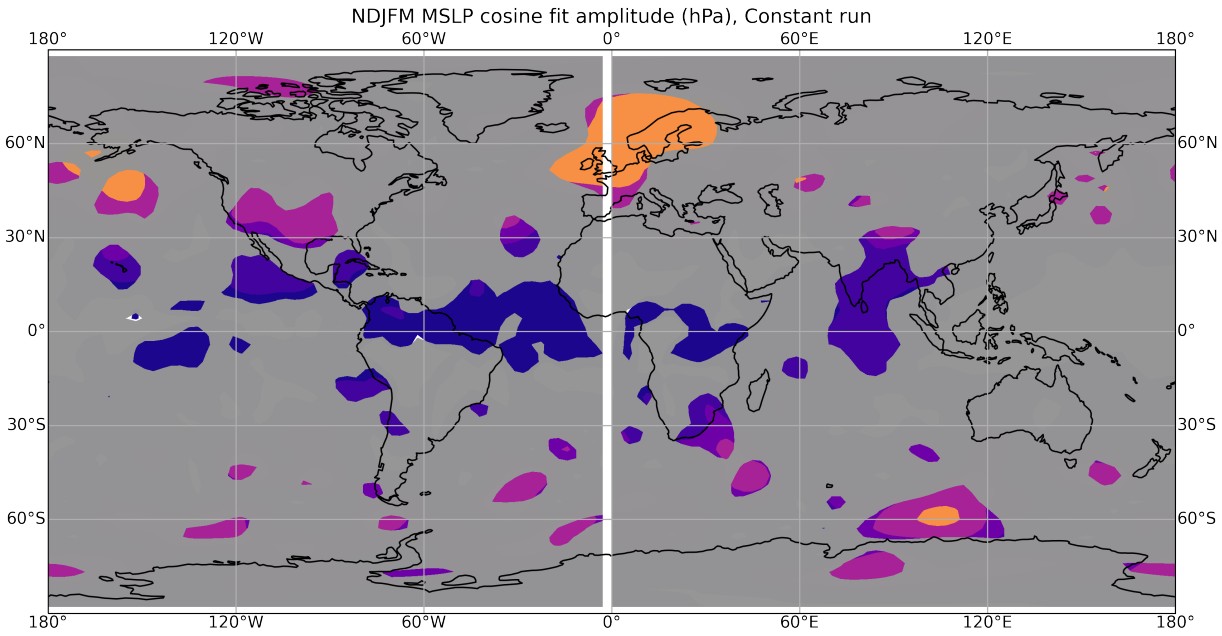

Figure 7. As Figure 5 but for November-March (NDJFM) mean sea level pressure anomaly (hPa).

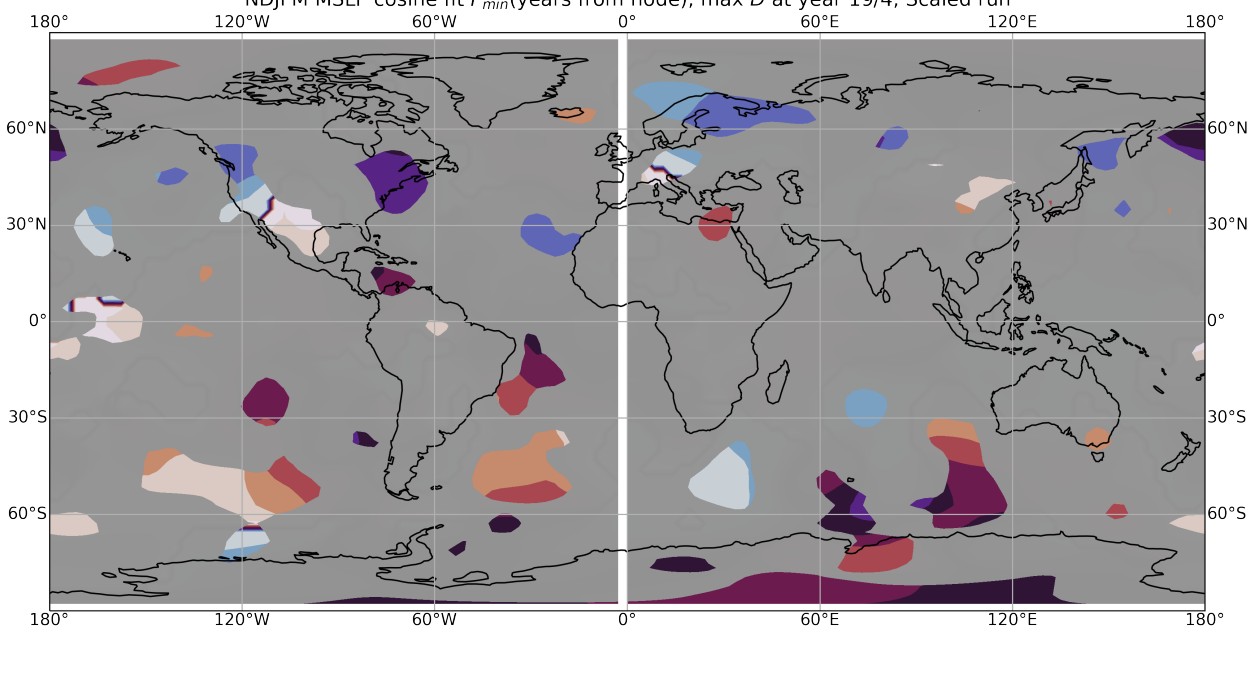

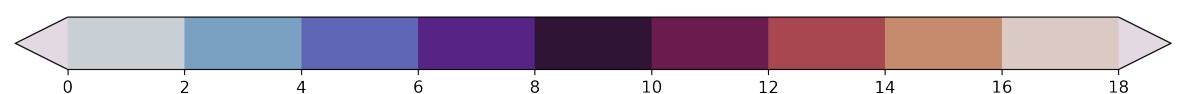

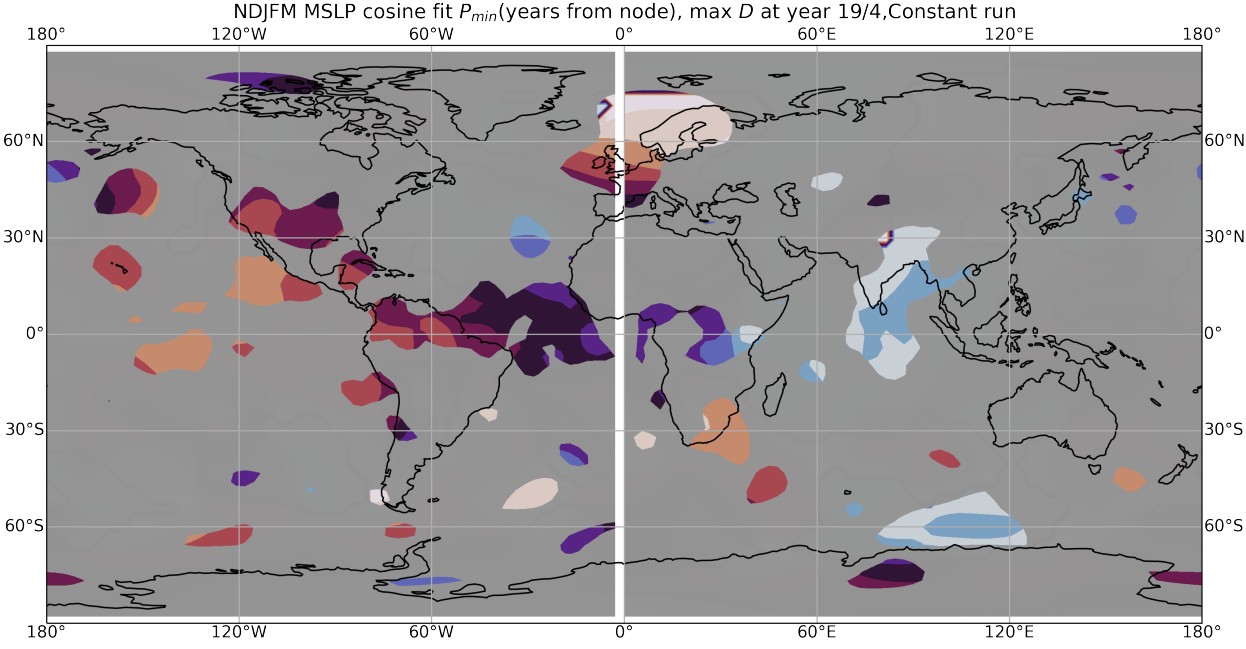

Figure 8. As Figure 6 but for November-March (NDJFM) mean sea level pressure anomaly (hPa).


There is some indication of a dipole across the Pacific Ocean, suggesting an El Niño Southern Oscillation (ENSO) response, which has been associated with the lunar nodal cycle (Loder and Garrett 1978, Yasuda 2018), but the response is not statistically significant. We have analysed surface temperatures in the Nino 3.4 region but find no significant signal. A similar

statistically insignificant result is found for the time-variation of the Atlantic Meridional Overturning Circulation (AMOC).
The size of these responses in the context of other forcings is discussed below.

The long-term effects of the lunar nodal cycle are now examined, since the effect of the ocean circulation would be expected to 'redden' a 19-year periodic forcing signal into lower frequencies, measurable on longer timescales. Figure 9 shows decadal $T_{surf}$ anomalies in each run. There is an increase in the standard deviation of running decadal-mean temperature in the CONSTANT run, but no clear increase in the SCALED run. The presence of the lunar nodal cycle will add a small positive or
negative tendency to warming decadal trends in the 21st century.

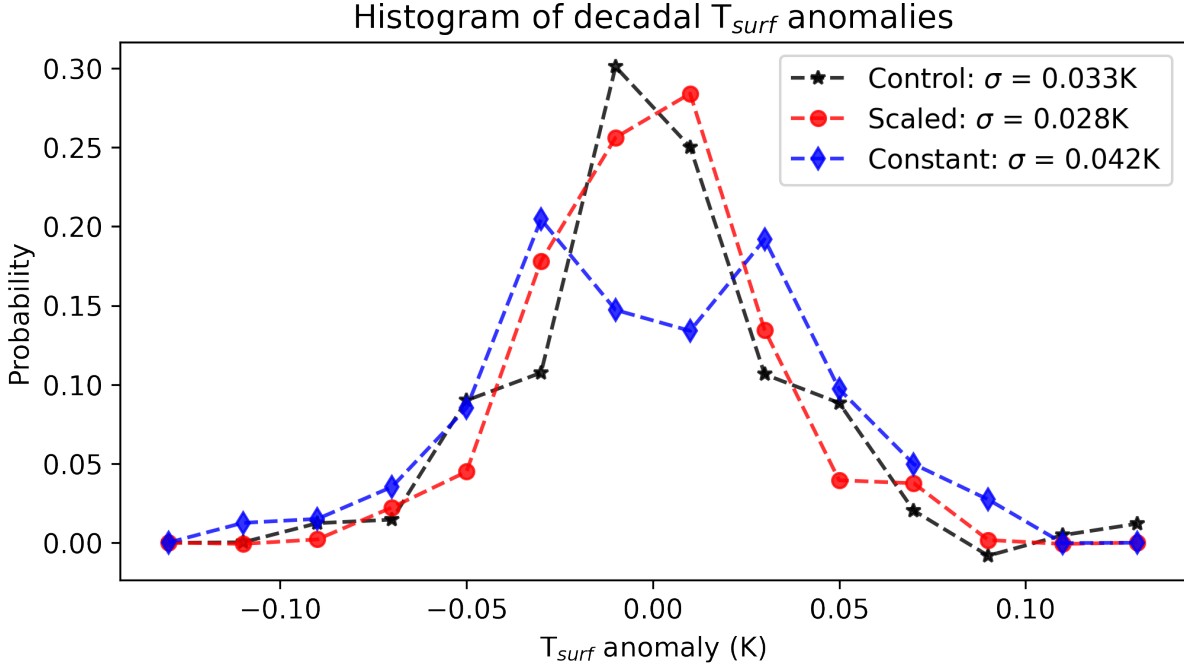

Figure 9. Histogram of decadally averaged surface temperature anomalies (K) in CONTROL (black), SCALED (red) and CONSTANT (blue)

Figure 10 (top panel) shows the assessed projections for global temperature for the five different emissions scenarios used in
the 6th Assessment Report of the Intergovernmental Panel on Climate Change (IPCC AR6; Lee et al. 2021). The assessed 5-95% uncertainty range is indicated with shading for the two more extreme scenarios (SSP5-8.5 and SSP1-1.9; the magnitude is similar for the other scenarios). To highlight the effect of the lunar nodal cycle on these assessed projections, we add a sinewave with peak amplitude of 0.04K, with the correct lunar nodal cycle timing, to each of the curves (bottom panel). The lunar nodal cycle is expected to act as a slight cooling influence on the climate in the mid 2020s, delaying the arrival of the
1.5C temperature threshold in SSPs with higher carbon emissions (shown in red), but is a warming influence in the early-mid 2030s, hastening the arrival of the 1.5C temperature threshold in SSPs with lower carbon emissions (shown in blue). The net effect is to reduce the spread in time at which the world is projected to reach 1.5C above pre-industrial levels from 5 years to 3 years.

**4. Discussion and Conclusions**

The timing of the lunar nodal cycle is of special interest when considering so-called hiatus and surge decades. A purported 'slowdown' of global temperatures at the start of the 21st century has been much discussed with mechanisms such as volcanic aerosol forcing (Santer et al. 2014) and stratospheric water vapour (Solomon et al. 2010) invoked as part of the explanation, although updated observational datasets show less of a global temperature slowdown than previously identified (e.g. HadCRUT5 and others). Anomalously high heat uptake by the world's oceans (Meehl et al. 2011, Guemas et al. 2013) and
circulation changes in the Pacific Ocean (Kosaka and Xie 2013) are also suggested. Figures 2 and 3 suggest a potential role

for the lunar nodal cycle in driving decadal variations in warming rates, with the SCALED run implying an average flux of ~0.07+/-0.07 Wm$^{-2}$ into the world's oceans over the period 2002-2011. While the uncertainty in the value is clearly large, its magnitude suggests that it cannot be discounted as a significant driver of multidecadal variability of global temperature, given that, for example, the additional heat uptake into the oceans through the surface during hiatus-type periods is approximately 0.7 Wm$^{-2}$ (Drijfhout et al. 2014).

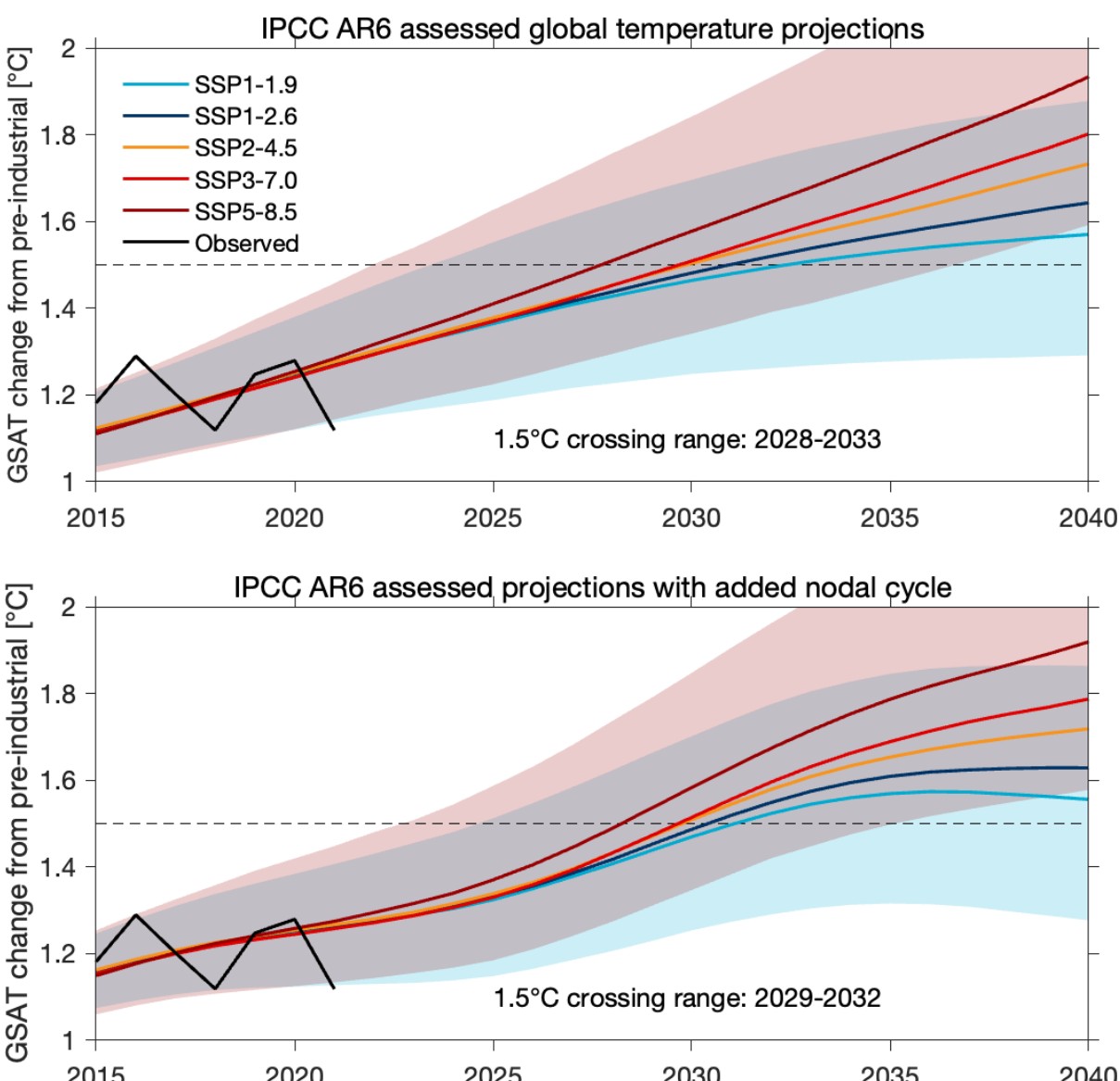

Figure 10. Top panel: IPCC AR6 assessed trends for different shared socioeconomic pathways (SSPs). Bottom panel: as top panel but with a lunar nodal cycle of 0.04K amplitude, chosen to be approximately an average of the CONSTANT and SCALED runs.

Figure 4 suggests that the contribution of the lunar nodal cycle should be a global cooling of 0.03-0.06 K over the period 2020-2029, and a warming of 0.03-0.06 K over the period 2030-2039. Given the magnitude of such changes, and the results shown in Figure 10, we suggest that a parameterisation of the lunar nodal cycle should be implemented in 1D integrated assessment models (IAMs) in order that they better represent the effect of this repeatable and predictable source of climate variability on the impacts of climate change. Although it is known that inclusion of the cycle affects projections of future regional sea-level

change, for example in the North Sea (Baart et al. 2012), we find that the global modulation of global sea level is 1-2 mm (not shown), because of counteracting influences of hot and cold anomalies in the ocean (Figure 2). Such a value is far less than the currently observed global sea rise of 3 mm yr$^{-1}$ (Dangendorf et al. 2019), suggesting that the impact of the lunar nodal cycle on global sea level rise is small.

The geographical response of the model to the lunar nodal forcing can be better understood by putting it in context with other modes of variability. Figure 5 shows that the response of the north Atlantic Ocean has an amplitude of order 0.1 K. For context, this is about 20-30% of the size of SST anomalies associated with Atlantic Multidecadal Variability (Omrani et al. 2022). The results shown in Figure 7 can be better understood by being put in the context of other sources of variation in the North Atlantic region. The natural variability of the NAO in FORTE2 and observations which has a peak-to-peak amplitude of 3 hPa and 4 hPa respectively (Blaker et al. 2021): the lunar nodal response is smaller, but certainly noticeable. For added context, the response of the NAO to observed Atlantic decadal SST variability is 2-3 hPa (Årthun et al. 2021), while the response to solar variability is 3-4 hPa (Gray et al. 2016), suggesting that the lunar nodal cycle has a much smaller, but noticeable effect on Atlantic European winter climate and the NAO.

The Arctic response is almost in antiphase with the rest of the world, reflecting the reversed temperature gradient in the upper ocean, i.e. the lack of a permanent stratifying thermocline. Maximum Arctic temperatures are modelled as occurring in years 5-9 of the cycle, which in reality correspond to 2007-2011, consistent with enhanced Arctic warming during this time (Stroeve et al. 2012). A caveat in interpreting the above results, as well as results suggesting a large response on the northern edge of Antarctica, is that the sea-ice representation of FORTE2 is simplified, consisting of one slab (Blaker et al. 2021). Future work regarding the nodal cycle in the subpolar and polar oceans should be carried out with a more realistic sea ice model, with other forcings included to assess potential nonlinear combinations of response. All other things being equal, similar warm Arctic anomalies might be expected during 2026-2030. Figure 7 also implies that the NAO is likely to be more negative than average at the same time.

A caveat in this work lies in the nature of the tracer vertical diffusion scheme which is being modulated. Here we use a simple profile that represents the sum of all diffusion processes that has been tuned to give a good representation of the global thermocline structure. State-of-the-art coupled climate models use a variety of more sophisticated vertical diffusion parameterisations in combination to represent a number of different processes, including wind mixing, tidal mixing, and internal gravity wave scattering (Mackinnon et al. 2017, de Lavergne et al. 2020). Only the tidal and internal tidal induced diffusion is enhanced by the lunar nodal cycle. Thus, if this accounts for one half of the mixing that we apply (a conservative estimate) then we would expect the magnitude of the response to be halved. An important point to note with regard to parameterisation of the nodal cycle is that tidal forcing is not necessarily at the same time and place where tidal dissipation takes place, implying a limit to the spatial resolution that a parameterisation might employ.

We have implemented a simple, flexible parameterisation of the lunar nodal cycle into an AOGCM, examined its effects on multicentennial-length runs, and have assessed its potential effects on 21st century climate. Our results lend further weight to the idea that the phenomenon should be parameterised in decadal-scale forecasts made using global circulation models (e.g. Osafune et al. 2014), as well as in integrated assessment models, given the potential effect of the lunar nodal cycle on future climatic trends.

**Author Contribution**

MJ conceived the original idea and performed model runs. All authors contributed to analysis of results and writing.

## Competing Interests

The authors declare that they have no conflict of interest

## Data Availability

Lunar nodal cycle forcing data available at:

https://research-portal.uea.ac.uk/en/datasets/lunar-nodal-cycle-amplitude-modulation-map

## Acknowledgements

The research presented in this paper was carried out on the High Performance Computing Cluster supported by the Research and Specialist Computing Support service at the University of East Anglia. MJ acknowledges the support of NERC project NE/N006348/1 (SMURPHS). We acknowledge useful discussions with Mark Prosser.

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

**Figure Legends**

Figure 1. Top panel- variation in time of the modulation (with reference year for illustrative purposes on the top axis). The tidal modulation f(x,y,t) in the model is the top panel f'(t) multiplied by the bottom panel f'(x,y), multiplied by 1.0 for run CONSTANT, and a scaled function in run SCALED. Bottom panel- geographical distribution of the modulation of tidally-driven diffusion by the 18.6 year lunar nodal cycle.

Figure 2. Top panel- globally averaged variation with phase of temperature anomalies vs ocean depth (K) vs tidal modulation phase in the SCALED run. Bottom panel- as top panel but for CONSTANT run.

Figure 3: Top panel- variation in time of the modulation. Middle panel – globally averaged ocean heat content anomaly ($10^{22}$ J) vs tidal modulation phase. Bottom panel- globally averaged surface ocean heat flux anomaly (W m$^{-2}$) vs tidal modulation phase. The mean +/- 2 standard errors are shown for CONSTANT in thin blue, and for SCALED in thin red; sinusoidal best fit curves of global temperature anomalies are shown for CONSTANT in thick blue, and for SCALED in thick red; the mean +/- 2 standard errors in the control integration of FORTE2 is shown for reference in in black.

Figure 4. As Figure 3 but Middle panel- globally averaged surface temperature vs tidal modulation phase. Bottom panel- as for middle panel but for surface temperature in the Arctic region (70°N-90°N).

Figure 5. Top panel- geographical variation in amplitude of sinusoidal trigonometrical fit to surface temperature (K) in SCALED; grey shaded areas show where amplitude is less than 2 standard errors in the CONTROL integration; note nonlinear contour interval. Bottom panel- as top panel but for CONSTANT.

Figure 6. Top panel- geographical variation in phase of sinusoidal trigonometrical fit to surface temperature (K) in SCALED. The value is the phase at which the fitted temperature reaches a minimum in terms of the year in the cycle; grey shaded areas show where amplitude is less than 2 standard errors in the CONTROL integration; Bottom panel- as top panel but for CONSTANT.

Figure 7. As Figure 5 but for November-March (NDJFM) mean sea level pressure anomaly (hPa).

Figure 8. As Figure 6 but for November-March (NDJFM) mean sea level pressure anomaly (hPa).

Figure 9. Histogram of decadally averaged surface temperature anomalies (K) in CONTROL (black), SCALED (red) and CONSTANT (blue)

Figure 10. Top panel: IPCC AR6 assessed trends for different shared socioeconomic pathways (SSPs). Bottom panel: as top panel but with a lunar nodal cycle of 0.04K amplitude, chosen to be approximately an average of the CONSTANT and SCALED runs.