# Peer review of "The modelled climatic response to the 18.6-year lunar nodal cycle and its role in decadal temperature trends"

_EGUsphere, 2022_

## Author Comment (AC1)

[revised manuscript text omitted]

310                                   **Figure 3**

[Figure]

**Figure 4**

315

[Figure]

[Figure]

**Figure 5**

[Figure]

[Figure]

320

**Figure 6**

[Figure]

[Figure]

325

**Figure 7**

[Figure]

[Figure]

330 **Figure 8**

[Figure]

335 **Figure 9**

[Figure]

**Figure 10**

---

## Author Comment (AC2)

**Response to reviewers (reviewers' comments in blue)**

**Review RC1:**

Two degrees is way too coarse to investigate tides. We're talking about a process that has a typical scale of fewer than 5 km at critical locations, and that is highly dependent on the correct representation of bathymetry.

The vertical diffusivity in the model is meant to represent a parameterisation of tidal and other mixing processes in the open ocean, rather than a parameterisation of tidal effects at very small coastal scales. Such a process is very widely used in ocean models. We have clarified the method to state that "all tidal energy is dissipated *at the 2° grid scale*.". See also Comment CC3 (Reply on RC1).

**2.The model's biases**
From line 111, you try and find an explanation for the seemingly "inconsistent" behavior of the Nordic Seas and later that of the Southern Ocean. From line 125, you report on potential but non-significant links with the modelled AMOC. Unfortunately, FORTE2, whose reference Blaker et al. (2020) was missing from the bibliography, has significant biases that most likely impact these two results:

- Warm bias in the Southern Ocean and in the northwest Atlantic;
- Deep mixed layer bias in the Nordic Seas and in the Ross Sea.

Similarly, you ought to verify the biases in the model's NAO before analyzing its response.

We thank the reviewer for spotting this unfortunate omission, and we have added the Blaker et al (2021) reference to the reference list. We note this is 2021, and not 2020, and have corrected the citation in the text.

The NAO and SAM (Southern Annular Mode) have been examined in Blaker et al (2021) (Figures 15 and 16 respectively) and the model does a good job of representing both. We have now changed the text to reflect this (new text in italics):

"The geographically varying phases suggest a potential for geographically varying temperature and circulation responses, *especially in the case of the North Atlantic Oscillation (NAO) and Southern Annular Mode (SAM), which are simulated quite well by FORTE2 (Blaker et al 2021)*."

**3.The focus on temperature only**
In regions that are salinity-controlled (Arctic, Nordic Seas, North Atlantic, Southern Ocean), the change in background diffusivity should primarily affect the salinity. A freshening would likely lead to a surface cooling, and a salinification to enhanced mixing so to a warming of the surface. Maps or timeseries of salinity changes should be shown. Similarly, there is currently no mention of sea ice changes in this paper.

We have now changed Figure 2 to include the variation of global salinity with lunar nodal cycle phase. We have examined Arctic temperature rather than sea-ice anomalies because of the relative simplicity of the sea-ice scheme in the model. We do now say in the discussion that future work should be conducted using a more complex sea-ice model:

*A caveat in interpreting the above results, as well as results suggesting a large response on the northern edge of Antarctica, is that the sea-ice representation of FORTE2 is simplified, consisting of one slab (Blaker et al. 2021). Future work regarding the nodal cycle in the subpolar and polar oceans should be carried out with a more realistic sea ice model, with other forcings included to assess potential nonlinear combinations of response.*

**Other comments:**
Inconsistent usage of "high latitudes". Line 69-70 for example, it excludes the Arctic (which is extremely stratified).

We have changed the text to say *Although the tidal modulation is largest (exceeding 10%) in the Arctic and Southern Oceans, high-latitude water columns are typically only weakly, or negatively, temperature stratified (i.e. the near-surface vertical gradient of temperature is either small or negative). So, counterintuitively, the effect of tidal modulation on climate in these regions might actually be small.*

Figures 5:8 are very hard to read. Use a pseudocolor plot, or at least filled contours.

We have changed Figures 5-8 to show filled contours for both data and contours for statistical significance to enhance readability. In particular Figures 6 and 8 have a cyclic colour scale for clarity. We have also corrected a mislabelling of the SSP Scenarios in Figure 10, and added model uncertainty for two future scenarios.

All figures are at the end of the document, and their captions are on a different page. It is really uncomfortable to view on a screen. The EGUsphere now (finally!) recommends that figures and their captions be in the text, closest to where they are discussed.

We have now moved all figures to the place in the text where they are discussed, and each figure is accompanied by its own caption.

---

## Author Comment (AC3)

**Response to reviewers (reviewers' comments in blue)**

**Review RC2:**

My impression is that the presented paper does the minimum necessary to draw attention to the potential importance of the 18.6 year lunar nodal cycle in the context of climate projections and hiatus/surge events. The authors propose that parameterisation of the lunar nodal cycle should be implemented in 1D integrated assessment models and decadal-scale forecast systems, and I am inclined to agree.

We thank the review for the above assessment.

I have some concerns that I would like to see addressed prior to publication.

**Major comments:**

The authors create a map of ocean diffusion amplitude modulation based on the geographical distribution of the RMS current velocity and the nodal amplitudes. However, these are the barotropic tides. Around 2/3 of the power input to surface tides is lost in the shallow seas, whilst the remaining 1/3 generate internal tides (see e.g. Ferrari and Wunsch, 2009; de Lavergne, 2019). I believe it is the latter which the authors intended to parameterise in the model, and I therefore have concerns about the spatial distribution given in Figure 1.

The geographical distribution of internal tidal energy dissipation is strongly influenced by bathymetry. The map of tidal dissipation produced by de Lavergne et al. (2018, 2019) clearly shows the influence of bathymetry. This prompts two questions:

- Why did the authors not use such a map in their parameterisation?
- How would the results differ if the dissipation used this sort of geographical distribution?

Such a change in the geographical distribution would likely affect many of the regional results, but it is harder to gauge the impact on the global quantities such as surface temperature and ocean heat uptake.

Our parameterisation of the 18.6-year lunar nodal cycle requires spatial fields for the eight largest tidal constituents. This is because the nodal amplitude is different for each tidal constituent (see table 1 in the revised manuscript).

Note that $S_2$ and $P_1$ are pure solar tides so are not directly modulated by the nodal cycle and that $M_2$ and $N_2$ are out of phase with the other constituents. So, although $M_2$, $K_1$ and $S_2$ are the most energetic tidal constituents globally. $K_1$, $O_1$ and $M_f$ are the constituents with the largest potential (i.e. nodal amplitude times typical magnitude) for 18.6-year modulation of tidally-driven vertical diffusivity.

Although we agree that much of the tidal forcing of deep ocean diffusivity is through the internal tide field, global maps of internal tide variability and dissipation are not available for all the key constituents required to do this sensitivity study. The global maps of internal tide generation and dissipation presented by de Lavergne et al. (2019; data doi: 10.17882/58105) only include the $M_2$, $S_2$, and $K_1$ constituents, plus an extrapolated 'all constituents' field. Similarly, the global maps of tidal mixing used by de Lavergne et al. (2020; data doi: 10.17882/73082) only contains constituent-integrated values.

To arrive at an appropriate spatial field to apply the 18.6-year modulation of tidally-driven vertical diffusivity, it is essential to use a consistent model for all the constituents. In the absence of a multiple-constituent global internal tide model, and given the relatively course 2° horizontal resolution of the ocean in our climate model, we rely on a barotropic tide model with the reasonable assumption that all tidal energy is dissipated locally. However, we acknowledge that some tidal energy does travel further than 2° through the internal tide field. Future work will use the global distribution of baroclinic tidal variability when such maps become available.

Figure 1 is the spatial distribution of the 18.6-year modulation of tidally-driven diffusivity that is applied to a conventionally horizontally uniform and temporarily constant vertical diffusivity. Thus, it cannot be compared with maps of direct tidal mixing.

I believe the importance of the result in the context of the recent hiatus in global temperature and ocean heat uptake is overstated. Hedemann et al. 2017 (cited on line 150) define an ocean surface layer that is 100m thick. Fluxes of heat into the ocean are given as fluxes through 100m, not the ocean surface, and are consequently much smaller. Estimates of increased ocean heat uptake (through the ocean surface) during the 2000s are typically 0.7 +/- 0.3 W m$^{-2}$ (Drijfhout et al. 2014). The average flux you report (~0.07 +/- 0.07 W m$^{-2}$) is therefore sufficient to explain one tenth of the hiatus.

This is a very good observation by the reviewer. Accordingly we have replaced the reference and changed the text to the following:

*While the uncertainty in the value is clearly large, its magnitude suggests that it cannot be discounted as a significant driver of multidecadal variability of global temperature, given that, for example, the additional heat uptake into the oceans through the surface during hiatus-type periods is approximately 0.7 Wm$^{-2}$ (Drijfhout et al. 2014).*

*Figure 4 suggests that the contribution of the lunar nodal cycle should be a global cooling of 0.03C-0.06C C over the period 2020-2029, and a warming of 0.03-0.06 C the period 2030-2039.*

See also our reply to review RC4 (1$^{st}$ major comment).

**Minor comments:**

Line 35: miss-spelt Yndestad.

We have corrected this error.

Line 89: remove "opposites" given in parenthesis to improve readability. They are unnecessary due to the last sentence in the paragraph.

Parentheses have been removed.

Line 98 and onwards: refers to "global mean surface temperature Tg", whilst the plot titles in Figure 4 refer to "Tsurf". It is ambiguous what "surface temperature" refers to. In the preceding paragraph I was (I think rightly) taking this to be the "sea surface temperature" (SST). However, I think this and subsequent references might be to "surface air temperature" (SAT; due e.g. to the presence of contours over land in figures 5 and 6). Please clarify throughout.

We have removed all references to $T_g$, and replaced with $T_{surf}$, as this is meant to refer to surface temperature (whether over land or ocean), and not SAT.

Line 102: please supply "(vol/sol refs here)".

This has been done with a reference to Gray et al.(2013) (also see response to Reviewer CC1).

Line 104: relating to my earlier comment, it is important to determine whether the quantity presented in Figure 4 is SST or SAT. If SAT then the contribution from the land will likely dominate the variability. If SST, does the variability arise from the summer months? In either case, I think a caveat drawing the reader's attention to the simple ice representation in FORTE2 would be advisable.

The quantity is surface temperature. FORTE2 does not have any model layers at 1-10m above the surface. We have added the following text as a caveat in the Discussion section:

*A caveat in interpreting these results is that the sea-ice representation of FORTE2 is simplified, consisting of one slab (Blaker et al. 2021). Future work regarding the nodal cycle in the Arctic should be carried out with a more realistic sea ice model.*

Line 110: remove 'though'

The word has been removed.

Line 111: Is the inconsistency in the Nordic Seas caused/dominated by variation in the ice cover, rather than the lunar tidal variation in the experiment?

The response is statistically significant in the Nordic Seas (see Figure 5), so is a response of the ice cover to the forcing. The caveat introduced into the discussion regarding the sea ice scheme (see above) should also hold for this result.

Line 120: Missing close ")"

Parenthesis closed

Line 125: switch order of the last two sentences in this paragraph.

The order has been switched.

Line 144: insert "a" > "…less of a global…"

"a" has been added.

Check references: missing Blaker et al. (2020)

The reference has been corrected to Blaker et al. (2021) and added.

Line 267/8: two mentions of "380 years" which seems to contradict the 760 years mentioned on line 80.

We have removed the wording. In addition we have corrected Figure 4 to note it is as Figure 3, not 2.

Line 279: duplicate "in in"

This has been removed.

---

## Author Comment (AC4)

**Response to reviewers (reviewers' comments in blue)**

**Review RC 4**
I find the topic interesting and I think effects like this should be explored further
We thank the reviewer for this assessment.

**Major comments:**
I think that discussion could be more thorough, i.e., results/discussion sections should be expanded.

- For example, how does lunar nodal cycle impact on global/regional mean temperature, NAO etc. compare with other processes that control decadal-multidecadal indices. Is it more or less important for climate system variability than other processes? Or perhaps the lunar nodal cycle is a cause for some of the variability? Maybe the different variabilities are out-of-phase and/or uncorrelated? Much like other comments I have seen, I agree that the results in this paper are overstated, also given the simplicity of the experiments.
  - In the Atlantic there is a 15–18-year cycle - see: Årthun, M., Wills, R. C. J., Johnson, H. L., Chafik, L., & Langehaug, H. R. (2021). Mechanisms of Decadal North Atlantic Climate Variability and Implications for the Recent Cold Anomaly, Journal of Climate, 34(9), 3421-3439
  - There are obviously also Pacific (inter-)Decadal variability, Atlantic Multidecadal variability, AMOC etc., which are briefly mentioned in the manuscript. See e.g.: Omrani, N.-E., et al., 2022: Coupled stratosphere-troposphere-Atlantic multidecadal oscillation and its importance for near-future climate projection. npj Clim. Atmos. Sci., 5:59
  - There are many more papers on the topic that could be further discussed.

This is a good point and we now discuss the size of the response in modes of variability in the context of other drivers in the text:

*The geographical response of the model to the lunar nodal forcing can be better understood by putting it in context with other modes of variability. Figure 5 shows that the response of the north Atlantic Ocean has an amplitude of order 0.1 K. For context, this is about 20-30% of the size of SST anomalies associated with Atlantic Multidecadal Variability (Omrani et al. 2022). The results shown in Figure 7 can be better understood by being put in the context of other sources of variation in the North Atlantic region. The natural variability of the NAO in FORTE2 and observations which has a peak to peak amplitude of 3 hPa and 4 hPa respectively (Blaker et al. 2021): the lunar nodal response is smaller, but certainly noticeable. For added context, the response of the NAO to observed Atlantic decadal SST variability is 2-3 hPa (Årthun et al. 2021), while the response to solar variability is 3-4 hPa (Gray et al. 2016), suggesting that the lunar nodal cycle has a much smaller, but noticeable effect on Atlantic European winter climate and the NAO.*
And
*Future work regarding the nodal cycle in the Arctic should be carried out with a more realistic sea ice model, with other forcings included in order to assess potential nonlinear combinations of response.*

See also our reply to reviewer RC2 (2$^{nd}$ major comment) regarding the results in the context of the global warming slowdown of the early 21$^{st}$ century.

- The authors state on l. 120, 125 there is insignificant response for everything, except maybe in MSLP in the Euro-Atlantic. How much variance in the NAO on this specific timescale does nodal cycle represent?

We now put the variability of the NAO in the context of other modes of variability and the mean variability in FORTE2:
*The natural variability of the NAO in FORTE2 and observations which has a peak-to-peak amplitude of 3 hPa and 4 hPa respectively (Blaker et al. 2021): the lunar nodal response is smaller, but certainly noticeable. For added context, the response of the NAO to observed Atlantic decadal SST variability is 2-3 hPa (Årthun et al. 2021), while the response to solar variability is 3-4 hPa (Gray et al. 2016), suggesting that the lunar nodal cycle has a much smaller, but noticeable effect on Atlantic European winter climate and the NAO.*

- L. 128-138: I think figures here need some uncertainty estimates. Also, I think this paragraph is overstated – other effects may be stronger than nodal cycle so I would like to caution against implying "nodal cycle will(has) cause(d) this". While I agree that decadal-multidecadal variability can cause delays in or speed-up the global warming trends (and affect the onset of 1.5 degree warming) I think you must be careful if you are not sure how much other modes of variability will contribute and to what extent – different effects may cancel out and then the statements in this paragraph are less meaningful.
    - Fig. 10: I am not sure how you added nodal cycle in for bottom panel in Fig. 10. Did you run the model? Statistically? Please elaborate.

The nodal cycle was added as a simple mathematical function. We now state this in the text:

*The assessed 5-95% uncertainty range is indicated with shading for the two more extreme scenarios (SSP5-8.5 and SSP1-1.9; the magnitude is similar for the other scenarios). To highlight the effect of the lunar nodal cycle on these assessed projections, we add a sinewave with peak amplitude of 0.04K, with the correct lunar nodal cycle timing, to each of the curves (bottom panel).*

  - Also add uncertainty from climate models on top panel.

We have added the uncertainty from two scenarios to Figure 10.

I think methods should be provided in more detail (use appendix if needed).
- I think that the authors have a control run, but it is never mentioned in the methods.

We have now clarified this in the Method section:

*FORTE2 is run for three configurations: pre-industrial control (as in Blaker et al. 2021), SCALED, and CONSTANT, for 2300 years, with years 1520-2280 being analysed, i.e. 760 years or 40 full cycles.*

- On l. 55 they talk about 8 largest tidal constituents – since I am not a tidal expert I find this hard to follow – please elaborate what they are, their timescales, is lunar nodal cycle among them or do you impose it separately (this seems to be the case).

We have included a new table (Table 1 in revised manuscript) highlighting the important characteristics of the eight tidal constituents we use in our parameterisation of the 18.6-year lunar nodal cycle. The lunar nodal cycle is not a specific tidal constituent, it is imposed by adding an 18.6-year oscillation to ocean vertical diffusivity with the spatial distribution shown in Figure 1. The spatial distribution is derived from TPXO7.2 modelled horizontal velocities for each constituent, along with their nodal amplitude. We have also modified the Method section to clarify this:

*The geographical shape of the function (Figure 1), determined by the relative strength of each tidal constituent at a given location and the constituent modulation amplitude, is multiplied by a normalised 18.6-year sinusoidal cycle to yield a spatially and temporarily varying modulation function. The phase of the modulation is such that, at most grid points, tidal currents are maximum at 4.75 years into the cycle (e.g., June 2006). The Pacific and Arctic Oceans…*

- On l. 65-70 you mention geographical shape of the function – is this based on observations? Which?

The geographic shape of the function is based on horizontal current velocities for eight tidal constituents from the TPXO7.2 inverse model and their nodal amplitudes. This is described within the revised Methods section (second paragraph).

- Presumably tidal components are typically parametrized in models?

In climate models, tidal components are generally not separated out, but their total effects are parameterised by a globally-averaged diffusivity that can vary in depth. Newer models are beginning to calculate the effects of e.g. bottom topography, but our wish was to create a parameterisation that was suitable for most global ocean models and AOGCMs in use today.

- On l. 71-77: authors talk about "SCALED" and "CONSTANT" model configurations and say that the former provides underestimations and the latter overestimation. Is there an ideal way of simulating this or are these methods commonly used – what have you simplified here?

There is no ideal way of simulating the vertical contribution of tides to the background diffusivity, so the method used seeks to give an upper and lower bound to the surface effects of tidal dissipation. We have now changed the wording to reflect this (new wording in italics):

*Given the uncertainties in the vertical contribution of tides to the background diffusion*, two *idealised* perturbation runs have been performed, one in which the nodal cycle parameterisation is applied uniformly with depth to the vertical diffusivity ("Constant"), and one in which it is applied such that its amplitude linearly decreases from 1 at a depth of 5000 m to 0 at the ocean surface ("Scaled"), to mirror the effect of tidal dissipation.

- L. 79: how exactly is nodal cycle applied to the model? Please elaborate.

We have now added this text to the method, and changed the caption of Figure 1 to be consistent with the new text, and referring to the new equation (1):

*The nodal cycle modulation is applied to the vertical diffusion with a period of 19 FORTE2 years, such that the total diffusion has the form has the form*

$$K' = K*T(t)*M(x,y)*S(z) \tag{1}$$

*where K is the standard background diffusion in FORTE2 (Blaker et al 2021), T(t) is the sinusoidal function of Figure 1 (top panel), M(x,y) is the geographically varying function in Figure 1 (bottom panel), and S(z) is unity for run CONSTANT, or the scaled function described above in run SCALED.*

Figures should have better captions – more descriptive – half of the time I am left wondering what is actually plotted. I also think they should be revised.

We have revised figure captions 1, 2, 3, 5, and 6 especially with regard to describing the phase of the lunar nodal cycle, and equation (1) above. Figures 5-8 are now filled contours, and Figures 6 and 8 have cyclic contour intervals for clarity- see also the reply to reviewer RC1.

- Fig. 2,3,4 it is really hard to see if something is out-of-phase/in-quadrature etc. if lines are plotted in different figures – I suggest plotting such lines together in one figure. Or provide more details – maybe Fig. references are incorrect in text or maybe you need to mention "middle panel in Fig. 3" etc.?

We have provided more details to clarify the figures, e.g. we have altered the discussion of Figures 2 and 3 to explain more clearly what we mean and explicitly refer to the phasing in terms of years (as in Figure 1 top panel):

Old text:

In both SCALED and CONSTANT cases, the top 100-150 m of ocean displays a cooling (warming) in phase with maximum (minimum) vertical diffusion. In the absence of any feedback from the atmosphere, the global mean sea surface temperature cold anomaly would be expected to peak half-way through the nodal cycle. However, the atmosphere almost immediately responds to the anomalously cool sea surface temperatures by fluxing heat into the ocean (Figure 3), causing an increase in total ocean heat content (Figure 3). The uptake of heat by the ocean results in a global ocean heat content anomaly approximately in quadrature with the surface heat flux and nodal cycle (Figure 2).

New text:

*In both SCALED and CONSTANT cases, the top 100-150 m of ocean displays a cooling in phase with maximum vertical diffusion in years 4-6. In the absence of any feedback from the atmosphere, the global mean sea surface temperature cold anomaly might be expected to peak half-way through the nodal cycle in years 9-10. However, as shown in Figure 3 (bottom panel), the atmosphere almost immediately responds to the anomalously cool sea surface temperatures by fluxing heat into the ocean during years 3-7, causing an increase in total ocean heat content between years 3 and 10 (Figure 3 middle panel). The uptake of heat by the ocean results in a global ocean heat content anomaly approximately in quadrature with the surface heat flux, i.e. maximum heat content is in years 9-10 (Figure 3 middle panel), while the maximum surface flux is at years 4-5, or approximately 4.5 years or 90º out of phase with the maximum heat flux.*

- l. 107-117: I cannot say I can follow the text here related to Figs. 5-6. I am not sure where you see out-of-phase relationship between Tsurf and global response (of what?).

- Fig. 7: Top panel does look NAO-like, but bottom panel reminds me more of blocking-like structure. Also, top panel shows perhaps some wave-trains in the Southern Hemisphere. I think this figure can be discussed more.

- Many figures are present, but not discussed enough – either don't use them or discuss them in more detail.

We have reworded and significantly lengthened the discussions of Figures 5-8. In particular, we have separated out the discussions of amplitudes and phases for clarity, and put more detail into the description. In addition Figures 5-8 have been replotted as shaded contour plots, with the phases being plotted using a cycling colour map, for added clarity. This section (lines 107-123 of the submitted manuscript) now forms lines 152-190 of the revised manuscript.

Is there any observational support for the authors' claims? Even if it is just 20 years of data (i.e. 1 cycle)?

The only regions with signals large enough to be seen over one cycle are where tides are very large, e.g. in shelf seas. We note such regions in section 1 of the manuscript (lines 34-35).

I agree with the authors' final statements that such effects (if they are as relevant as the authors claim) should be better represented in climate models.

We thank the reviewer for this assessment.

**Minor comments**

l. 17: O (0.1K) – are you trying to say that it is of order 0.1K? Then just spell it out.

We have made this change.

l. 32: 3.7% and 11.5% - provide reference for the numbers.

The 3.7% modulation amplitude for $M_2$ and 11.5% for $K_1$ come directly from the nodal amplitudes now presented in Table 1 and are referenced to Pugh (1987).

l. 42, 174: OAGCM --> AOGCM (?)

We have replaced OAGCM with AOGCM everywhere in the text

l. 98, 99: Tg – is this supposed to be Tsurf? It is not defined anywhere.

We have removed all references to Tg in the text

l. 100-102: suddenly you talk about solar/volcanic forcing – where is this from?? Reference figure/previous study.

We now reference a previous study on the impacts of the 11-year solar cycle on climate.

l. 106: 'later' --> 'below' (?)

We have made this change.

l.269: I think top and bottom panel description is reversed.

We have altered the caption to correct this, and give more detail (see reply to L. 79 comment above).

Fig. 2 caption: Provide units.

The units (K) are in the caption, but we have expanded each caption to describe both tidal modulation and temperature response.

All Fig. captions: more details.

---

## Author Response (AR2)

Dear Editor-

We have amended the manuscript in accordance with the wishes of the reviewer. Please find our specific comments below (reviewer's comments are in blue):

*(Note from editor)* "the tidal forcing is not necessarily the time and place where tidal dissipation takes place. This certainly adds a complication to the interpretation, and I think it would be beneficial to include this point in the discussion."

We have now added this text to the discussion (lines 283-285 of revised manuscript): *An important point to note with regard to parameterisation of the nodal cycle is that tidal forcing is not necessarily at the same time and place where tidal dissipation takes place, implying a limit to the spatial resolution that a parameterisation might employ.*

Minor comment:
Line 114: I'm not sure why the SST cold anomaly might be expected to peak halfway through the nodal cycle. Can you explain? I would expect the SST to respond more or less in phase with the cycle (with maybe a ~1 year lag due to reemergence-like behaviour). The surface layer is well mixed, so why should SST act differently to the top 100-150 m (line 113)?

We have changed the text to clarify this (lines 117-120 of revised manuscript): *In the absence of any feedback from the atmosphere, the global mean sea surface temperature cold anomaly might be expected to peak at the same time as the subsurface warm anomaly, which is half-way through the nodal cycle in years 9-10. This is when the tidally driven diffusion changes from its enhanced phase to a reduced phase.*

Typos:
Line 41: "coupled ocean-atmosphere" needs to become "coupled atmosphere-ocean" to match the acronym "AOGCM".
Line 71: "temporarily" > "temporally"
Line 131: I think "the maximum heat flux" is meant to be "the maximum heat content" on this line.
Line 193: "North east" > "North East"
Line 213: Missing "overturning" from "Atlantic meridional overturning circulation"
Line 250: Missing "over" from "warming of 0.03-0.06 K over the period"

We have made all these changes

Check throughout:
"et al" is an abbreviation, so should be "et al."
There are several instances of "Tsurf" where the "surf" is not subscript.

We have corrected all instances of this

Figures:
The figures are very low resolution. I cannot read most of the text in Figure 2. The rest are readable, but blurry. Please check the final figure quality carefully.

We have remade the paper figures using PDFs to be less blurry